# Surface Plasmon Resonance Biosensors with Magnetic Sandwich Hybrids for Signal Amplification

**DOI:** 10.3390/bios12080554

**Published:** 2022-07-22

**Authors:** Ting Sun, Mengyao Li, Feng Zhao, Lin Liu

**Affiliations:** 1Key Laboratory of Functional Organic Molecule, School of Chemistry and Materials Science, Guizhou Integrated Research Center of Polymer Electromagnetic Materials, Guizhou Education University, Guiyang 550018, China; sunt725726@163.com; 2College of Chemistry and Chemical Engineering, Anyang Normal University, Anyang 455000, China; 19837235165@163.com

**Keywords:** surface plasmon resonance, biosensors, magnetic preconcentration, signal amplification

## Abstract

The conventional signal amplification strategies for surface plasmon resonance (SPR) biosensors involve the immobilization of receptors, the capture of target analytes and their recognition by signal reporters. Such strategies work at the expense of simplicity, rapidity and real-time measurement of SPR biosensors. Herein, we proposed a one-step, real-time method for the design of SPR biosensors by integrating magnetic preconcentration and separation. The target analytes were captured by the receptor-modified magnetic nanoparticles (MNPs), and then the biotinylated recognition elements were attached to the analyte-bound MNPs to form a sandwich structure. The sandwich hybrids were directly delivered to the neutravidin-modified SPR fluidic channel. The MNPs hybrids were captured by the chip through the neutravidin–biotin interaction, resulting in an enhanced SPR signal. Two SPR biosensors have been constructed for the detection of target DNA and beta-amyloid peptides with high sensitivity and selectivity. This work, integrating the advantages of one-step, real-time detection, multiple signal amplification and magnetic preconcentration, should be valuable for the detection of small molecules and ultra-low concentrations of analytes.

## 1. Introduction

Surface plasmon resonance (SPR) biosensors have been used for the real-time detection of different classes of biomarkers with the advantages of simple operation, rapid response and high selectivity. However, traditional SPR sensors are less sensitive for the direct detection of small molecules (<8 kDa) or ultra-low concentrations (1 pM) of analytes based on the change in refractive index or thickness near metal surfaces [1,2]. In view of this, many efforts have been made to improve the sensitivity of SPR biosensors using secondary signal amplification with gold nanomaterials, magnetic nanoparticles and biomacromolecules [3,4,5,6,7,8,9,10,11,12]. The targets can be determined with a competitive (signal-off) or sandwich-like (signal-on) format. In contrast to the signal-off assay, the signal-on sandwich assay exhibits higher sensitivity because of the low background signal [13]. In this method, antibodies or other receptors are usually anchored on the surface of an SPR chip for the capture of target analytes [14,15]. Then, the recognition element-modified biomacromolecules or nanomaterials with large molecular weight, large size or high refractive index are delivered onto the chip’s surface to amplify the SPR signal. The secondary signal amplification can improve the sensitivity and avoid the influence of other components in biological fluids. However, these advantages come at the expense of simplicity, rapidity and real-time measurement due to a greater number of steps in the SPR analysis process [13]. Moreover, the storage of receptor-modified chips requires harsh conditions because of the limited stability of receptors such as antibodies under non-physiological conditions and high temperature [16]. Thus, it is of great significance to develop a general SPR biosensor for the detection of small molecules and ultra-low concentrations of analytes by integrating the advantages of one-step, real-time detection and multiple signal amplification.

Magnetic nanoparticles (MNPs) provide a feasible solution to the persistent challenges arising from the sensitivity and nonspecific adsorption of the preconcentration and separation of target analytes. Recently, some innovative sensing concepts involving MNP technologies have been reported for the ultra-sensitive detection of cells, nucleic acids, proteins and small biomolecules [17]. Usually, the target analytes are captured by the MNPs modified with specific receptors and then collected with a magnetic field. The analyte-bound MNPs can be brought to the sensor surface or can be removed from the sample and re-dispersed in solution. A good example of the concept is the electrochemical magnetobiosensor [18,19,20], in which the analyte-bound MNPs are collected by a magnetic electrode to produce a strong electrochemical signal. Alternately, the MNPs can also be captured by nucleic acid- or antibody-modified solid surfaces to produce optical signals [6,7,21,22,23,24,25]. For instance, MNPs have been successfully used to enhance the SPR signal due to their advantages of high molecular mass, high refractive index, low production cost and easy synthesis through hydrothermal and co-precipitative methods [4,26,27,28,29,30,31,32]. However, there are still some challenges related to the real-time SPR assays using MNP technologies [33]. For instance, the chips need to be covered with specific receptors through appropriate modification methods and the storage of the sensor chips requires harsh conditions. The capture of analyte-bound MNPs through the hybridization reaction or antibody–antigen interaction requires a long hybridization time and a low flow rate. Moreover, the immobilization of receptors on the chip surface may produce a steric hindrance for the capture of analyte-bound MNPs at the solid–liquid interface [34]. Therefore, new MNP-based SPR biosensors are still desired to simplify the detection procedure, shorten the analysis time and improve the detection efficiency.

Apart from the hybridization reaction and antibody–antigen interaction, the avidin-biotin system has been extensively used in biomedical analysis. Avidin and its analogues, including streptavidin (SA) or neutravidin (NA), modified on a solid surface show high stability, easy controllability and excellent flexibility [35,36,37]. Many SA or NA-labeled signal tags or solid supports and biotinylated reagents are commercially available. Herein, we proposed an MNP-based strategy for the design of one-step, real-time SPR biosensors based on the avidin–biotin interaction. As shown in Figure 1, the MNPs were modified with the receptors such as DNA or antibodies for the preconcentration and separation of target analytes through the hybridization reaction or antibody–antigen interaction. Then, the biotinylated recognition elements (DNA or second antibody) were attached onto the MNPs’ surface by binding to the targets via the same interactions. The NA-covered chips were utilized to capture the sandwich hybrids formed between biotinylated recognition elements, analytes and receptor-modified MNPs. The attachment of such sandwich hybrids is dependent upon the NA–biotin interaction, and the number of hybrids is proportional to the concentration of targets. Thus, the analysis of targets was converted to the direct detection of sandwich hybrids, significantly amplifying the signal due to the large size of MNPs and antibodies. Moreover, the magnetic preconcentration and separation can eliminate the nonspecific adsorption of other components in biological samples onto the chip surface, thus remarkably improving the sensitivity and selectivity. The dissociation constant between NA and biotin is in the order of 10^14^ M^−1^. Such a powerful and specific interaction will facilitate the capture of the hybrids at a faster flow rate, thereby shortening the analysis time and decreasing the nonspecific interactions.

## 2. Materials and Methods

### 2.1. Chemicals and Materials

Carboxylated MNPs with a diameter of ~100 nm were ordered from Ruixi Biotech. Co., Ltd. (Xi’an, China). Free and FITC-labeled NA proteins were provided by Thermo Fisher Scientific (Shanghai, China). DNA, avidin and glutathione (GSH) were obtained from Sangon Biotech. Co., Ltd. (Shanghai, China). Beta-amyloid peptides, human serum protein (HSA), beta-secretase, bovine serum protein (BSA), 1-ethyl-3-[3-dimethylaminopropyl]carbodiimide hydrochloride (EDC), N-hydroxysulfosuccinimide (NHS) and 1,1,1,3,3,3-hexafluoro-2-propanol (HFIP) were acquired from Sigma-Aldrich (Shanghai, China). The sequences of DNA are 5′-NH_2_-TTT TTG TAA AAC GAC GGC CAG-3′ (capture probe DNA), 5′-TAG GAA TAG TTA TAA CTG GCC GTC GTT TTA C-3′ (target DNA, T_DNA_) and 5′-TTA TAA CTA TTC CTA-biotin-3′ (bio-DNA). Monoclonal antibody (Ab_1_) and biotinylated second antibody (bio-Ab_2_) specific to Aβ_40_ were obtained from Covance Inc. (Dedham, MA, USA). Aβ_16_ was provided by China Peptide Co., Ltd. (Shanghai, China). Other reagents were ordered from Aladdin Reagent Co., Ltd. (Shanghai, China). The peptide of Aβ_40_ was dissolved in HFIP solution and then reconstituted in NaOH to a concentration of 1 mM. Before its use, the peptide was diluted with 10 mM phosphate buffer (pH 7.2) to a desired concentration.

### 2.2. Functionalization of MNPs

Carboxylated MNPs were modified with capture probe DNA or antibody (Ab_1_) by the EDC/NHS-activated covalent coupling reaction. Briefly, 5 mg of MNPs was dispersed in 2 mL of phosphate buffer solution containing 50 mM EDC and 10 mM NHS for 15 min. After the carboxylic groups on the nanoparticle surface were activated, the MNPs were washed with water and re-dispersed in 2 mL of phosphate buffer solution containing 0.2 μM NH_2_-DNA or 50 μg/mL of Ab_1_. After incubation for 2 h, the unreacted activated carboxylic groups on MNPs were blocked by incubation with 10 mM ethanolamine for 30 min. After being rinsed with water to remove the unreacted substances, the resulting DNA-MNPs or Ab_1_-MNPs were dispersed in 2 mL of phosphate buffer containing 10 μM BSA and stored at 4 °C for use. The change in the zeta potential of MNPs induced by surface modification was monitored on a zeta potential analyzer (Nano ZS, Malvern Company, Worcestershire, UK).

### 2.3. Preparation of SPR Chips

The gold chips were cleaned by a hydrogen flame and then incubated with 1 μM NA in carbonate buffer (pH 10) for 2 h. NA proteins were capped on the gold surface through the hydrophobic and Au–S interactions [38]. After washing the chip with buffer to remove the unbound NA proteins, the unreacted gold surface was sealed by incubation of the chip with 10 μM BSA and 100 μM GSH for 30 min. The resulting NA/BSA/GSH-covered chips were thoroughly washed and stored at 4 °C for use.

### 2.4. SPR Analysis

Before the SPR assay, 900 μL of target DNA (T_DNA_) or Aβ_40_ sample at a given concentration was spiked with 50 μL of 5 nM biotinylated DNA (bio-DNA) or bio-Ab_2_ at 37 °C. Then, 50 μL of DNA-MNPs or Ab_1_-MNPs suspension was added to the sample. After incubation for 1 h, the MNPs were rinsed three times with a magnetic field and then dispersed in 0.5 mL of phosphate buffer. For the real-time measurement, the NA/BSA/GSH-covered chip was fixed on a BI-SPR 3000 instrument (Biosensing Instrument Inc., Tempe, AZ, USA). When a stable baseline was attained, the resulting bio-DNA/T_DNA_/DNA-MNPs or bio-Ab_2_/Aβ_40_/Ab_1_-MNPs hybrids were delivered to the fluidic channel by a syringe pump, and the signal was recorded according to the change in the SPR refractive index.

## 3. Results and Discussion

### 3.1. Principle and Feasibility

To demonstrate the practicability of the strategy, a DNA biosensor was constructed by the formation of sandwich bio-DNA/T_DNA_/DNA-MNPs hybrids. The principle of the method was depicted in Figure 1A. The capture probe DNA was immobilized on the surface of MNPs to generate DNA-MNPs. When the T_DNA_ was hybridized with bio-DNA and the capture probe DNA on MNPs, the sandwich bio-DNA/T_DNA_/DNA-MNPs hybrids were formed. The excessive bio-DNA and other substances in the samples were then removed, and the T_DNA_ targets were concentrated with a magnetic field. The re-dispersed sandwich hybrids were then delivered to the fluidic channel and captured by NA proteins modified on the chip, thus producing an enhanced SPR signal due to the large molecular mass and high refractive index of MNPs. In this work, NA-covered gold chips were used to capture the biotinylated probe. To prove that NA can be anchored on the gold surface, the chip incubated with FITC–NA instead of NA was characterized by fluorescence microscope. As shown in Figure 1A, a green, fluorescent image was observed on the chip coated with FITC–NA/BSA/GSH, while the BSA/GSH-covered chip was black. This is indicative of the successful attachment of NA proteins on the chip surface. Moreover, the successful modification of DNA onto the surface of MNPs was confirmed by monitoring the change in the zeta potential. The zeta potential of MNPs changed from −11.2 to −13.7 mV after the modification of the capture probe DNA. After the hybridization reaction, the zeta potential became −18.2 mV. The increase in the negative charge is indicative of the formation of bio-DNA/T_DNA_/DNA-MNPs hybrids.

As expected, injection of the bio-DNA/T_DNA_/DNA-MNPs hybrids into the sensor channel caused a significant change in the SPR dip shift (curve a in Figure 1B). However, a negligible change was observed when injecting DNA-MNPs (curve b) or T_DNA_/DNA-MNPs (curve c) to the channel. In the absence of T_DNA_, the signal was close to the background level, indicating that there is no nonspecific adsorption between DNA-MNPs and the sensor chip. A control experiment was conducted by injecting the bio-DNA/T_DNA_/DNA-MNPs hybrids into BSA/GSH instead of the NA/BSA/GSH-modified chip (curve d). Consequently, no significant change in the SPR dip shift was observed. Therefore, the change in curve a can be attributed to the T_DNA_-triggered hybridization reaction and the NA–biotin interaction. In addition, a smaller SPR dip shift was attained when injecting bio-DNA/T_DNA_/DNA hybrids into the NA/BSA/GSH-covered chip (curve e), indicating that the SPR signal can be greatly intensified by MNPs (curve a).

### 3.2. Sensitivity for DNA Detection

After the successful preparation of the SPR biosensor, T_DNA_ at different concentrations was analyzed. Figure 2A shows the SPR curves obtained when the DNA-MNPs and bio-DNA were incubated with different concentrations of T_DNA_. The SPR dip shift increased with the increase in T_DNA_ concentration, further demonstrating that the signal change resulted from the binding of sandwich hybrids. A good linear relationship between the T_DNA_ concentration and the SPR dip shift was attained (Figure 2B). The linear equation can be expressed as θ = 0.10[T_DNA_] (fM) + 6.54 in the concentration range of 1–2000 fM (the inset). The high sensitivity with a detection limit of 1 fM can be attributed to the signal amplification of MNPs and the magnetic preconcentration of T_DNA_. Moreover, since the hybridization reaction was performed in the solution and the NA–biotin system shows a strong interaction, the real-time SPR assays have been achieved at a fast flow rate, shortening the analysis time and decreasing the nonspecific interactions.

### 3.3. Immunoassays of Aβ_40_

Immunoassays with the signal amplification of nanomaterials, including MNPs, have been widely applied in clinical analysis, food safety control, environmental monitoring and so on [39,40]. To demonstrate the applications of the sensing strategy in different fields, an SPR immunosensor was constructed and used for the detection of amyloid peptide Aβ_40_. As shown in Figure 1B, the Ab_1_-MNPs were used to capture Aβ_40_ and bio-Ab_2_ through the specific antigen–antibody recognition. After the formation of sandwich bio-Ab_2_/Aβ_40_/Ab_1_-MNPs, the hybrids were separated and then delivered to the NA/BSA/GSH-covered chip. As shown in Figure 3, a negligible change in the SPR dip shift was observed when injecting Ab_1_-MNPs (curve a) or Aβ_40_/Ab_1_-MNPs (curve b) into the sensor chip. However, a significant change was attained after injection of bio-Ab_2_/Aβ_40_/Ab_1_-MNPs to the chip surface (curve c). The value is much higher than that by injecting bio-Ab_2_/Aβ_40_/Ab_1_ into the chip (curve d), indicating that MNPs indeed enhanced the SPR signal due to the high molecular mass and high refractive index.

The analytical performance for the detection of Aβ_40_ was investigated by determining different concentrations of Aβ_40_. As shown in Figure 4A, the SPR dip shift increased gradually when the Ab_1_-MNPs were incubated with increasing concentrations of Aβ_40_ in the presence of bio-Ab_2_, indicating that the formation of sandwich hybrids was dependent upon the concentration of Aβ_40_. The method allowed for the detection of Aβ_40_ with a linear range of 10–2500 fM (Figure 4B). The linear equation between the Aβ_40_ concentration and the SPR dip shift was found to be θ = 0.11[Aβ_40_] (fM) + 3.34. The detection limit was estimated to be 5 fM, which is lower than that achieved by other methods (Table 1). The high sensitivity can be attributed to the multiple signal amplification by antibodies and MNPs and the magnetic preconcentration. Thus, the one-step, real-time assay was particularly suitable for the detection of small molecules and ultra-low concentrations of analytes. Plasmonic nanoparticles (e.g., Au and Ag) exhibit strong optical resonance for visible and near-infrared wavelengths. We believe that the analytical performance, including detection limit and linear range, could be further improved by using magneto-plasmonic nanostructures as the signal labels [23,26,27,28,41].

### 3.4. Selectivity for Aβ_40_ Detection

In addition to the high sensitivity, the specific antigen–antibody recognition promised excellent selectivity of the biosensor. Figure 5 shows the results for analysis of different samples, including other Aβ fragments (Aβ_16_ and Aβ_42_), serum protein HSA, beta-secretase and biotin-binding protein avidin. When the Ab_1_-MNPs and bio-Ab_2_ were incubated with Aβ_16_, Aβ_42_, HSA, beta-secretase and avidin, even at a higher concentration, no significant change in the SPR dip shift was observed (bars 1–5). The result indicated that the biosensor exhibited high selectivity. We also found that the coexistence of avidin in the Aβ_40_ sample decreased the SPR signal for the detection of Aβ_40_ (cf. bars 6 and 7). The result is understandable since avidin can bind to bio-Ab_2_/Aβ_40_/Ab_1_-MNPs through the avidin–biotin interaction, thus preventing the attachment of the hybrids to the chip’s surface. However, the interference can be readily eliminated by two-step magnetic separation. As expected, a high SPR signal was attained when the Ab_1_-MNPs were incubated with the mixture of Aβ_40_ and avidin and then separated by a magnetic field, followed by incubation of the Aβ_40_/Ab_1_-MNPs with bio-Ab_2_ and separated by a magnetic field again (bar 8). The result not only demonstrated that the signal was enhanced by MNPs, but it also indicated that the capture of the Ab_1_-MNPs was dependent upon the antigen–antibody and NA–biotin interactions. Moreover, the high sensitivity and magnetic preconcentration are conducive to improving the selectivity because the interference from other components in complex, real samples can be decreased by sample dilution and magnetic separation. Thus, the SPR immunosensor should be an excellent candidate for the detection of MNPs-based extracted products.

## 4. Conclusions

In summary, we proposed a general strategy for the design of SPR biosensors by integration of magnetic preconcentration. The signal was amplified by the integrated sandwich hybrids composed of recognition elements, target analytes, receptors and MNPs. The performance of the method was evaluated by detecting DNA and Aβ_40_ in which the sandwich hybrids were directly delivered to the sensor surface. The target concentration, down to 1 fM for DNA or 10 fM for Aβ_40_, can be readily measured. The method exhibits the merits of rapid response, real-time measurement, high sensitivity and excellent specificity. Since SA and NA proteins have high thermal stability and excellent biocompatibility and many biotinylated recognition elements are commercially available, the method would explore new applications of SPR biosensors, especially for the detection of small molecules and ultra-low concentrations of analytes. Moreover, novel electrochemical and optical technologies can be developed based on the magnetic preconcentration and avidin–biotin interaction.

## Data Availability

Not applicable.

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
