# Peer review of "Surface Plasmon Resonance Biosensors with Magnetic Sandwich Hybrids for Signal Amplification"

_biosensors, 2022, doi:10.3390/bios12080554_

Round 1
Reviewer 1 Report
The manuscript discussed the case in which an amplification technique using receptor-linked functionalized magnetic nanoparticles for significant improvement of surface plasmon resonance biosensor. And the relative results demonstrated a remarkable synergistic effect between receptor-linked functionalized magnetic nanoparticles and avidin-biotin interaction in improving separation/preconcentration. However, a major revision to this manuscript was requested. Authors should make major revision to this manuscript to improve their manuscript.
1. There were so many previous reports regarding to using receptor-linked magnetic nanoparticles for significant improvement of surface plasmon resonance biosensor. The originality of this paper was not so strong. The authors should provide detailed explanation for the differences in originality with previous reports.
2. Authors should provide the synthetic details in the Experimental Section (especially for the synthesis of magnetic nanoparticle).
3. Authors should provide the high-resolution fluorescence microscopy image of the FITC-NA/BSA/GSH in Figure 1A, which confirm that NA can be anchored on the gold surface.
4. Authors should provide the comparison of the UV–vis spectra (including MNPs, DNA, Ab1, DNA(Ab1)-MNPs, TDNA(target)/DNA(Ab1)-MNPs, bio-DNA(bio-Ab2)/TDNA(target)/DNA(Ab1)-MNPs) to show the successful vaccination of receptor and functionalized magnetic nanoparticles.
5. Authors should provide the comparison of the SPR sensorgrams (including Ab1-MNPs, target/Ab1-MNPs, bio-Ab2/target/Ab1, and bio-Ab2/target/Ab1-MNPs) to show the successful formation of the immunocomplex.
6. In Figure 2(B), authors should add two additional concentrations (in the range between 500~2000 fM) of TDNA to demonstrate practicality of the biosensor in linear quantification range.
7. Please review the language and representation of the whole manuscript to be clear, precise, accurate and focused.
Author Response
We thank the reviewer for his/her positive and constructive comments: “The manuscript discussed the case in which an amplification technique using receptor-linked functionalized magnetic nanoparticles for significant improvement of surface plasmon resonance biosensor. And the relative results demonstrated a remarkable synergistic effect between receptor-linked functionalized magnetic nanoparticles and avidin-biotin interaction in improving separation/preconcentration. However, a major revision to this manuscript was requested. Authors should make major revision to this manuscript to improve their manuscript.”
Comment 1: “There were so many previous reports regarding to using receptor-linked magnetic nanoparticles for significant improvement of surface plasmon resonance biosensor. The originality of this paper was not so strong. The authors should provide detailed explanation for the differences in originality with previous reports.”
Response: In the previous reports, antibodies or other receptors are usually anchored on the surface of SPR chip for the capture of target analytes, and then the recognition element-modified magnetic nanoparticles with large molecular weight and high refractive index are delivered onto the chip surface to amplify the SPR signal. The secondary signal amplification can improve the sensitivity and avoid the influence of other components in biological fluids. However, these advantages come at the expense of simplicity, rapidity and real-time measurement due to a greater number of steps in the SPR analysis process. Moreover, the storage of receptors-modified chips requires harsh conditions because of the limited stability of receptors such as antibodies under non-physiological conditions and high temperature, the capture of analytebound MNPs through the hybridization reaction or antibody-antigen interaction requires a long hybridization time and a low flow rate, and the immobilization of receptors on the chip surface may produce a steric hindrance for the capture of analytebound MNPs at the solid-liquid interface. Thus, it is of great significance to develop a general SPR biosensor for the detection of small molecules and ultra-low concentration of analytes by integrating the advantages of one-step real-time detection and multiple signal amplification. Avidin and its analogues including streptavidin (SA) or neutravidin (NA) modified on the solid surface shows high stability, easy controllability and excellent flexibility. Many SA or NA-labeled signal tags or solid supports and biotinylated reagents are commercially available. Herein, we proposed a one-step real-time method for the design of SPR biosensors by integration of magnetic preconcentration and separation. The targets were captured by the receptor-modified magnetic nanoparticles (MNPs) and then the biotinylated recognition elements were attached on the analytebound MNPs to form a sandwich structure. The sandwich hybrids were directly delivered to the neutravidin-modified SPR fluidic channel. The MNPs hybrids were captured by the chip through the neutravidin-biotin interaction, resulting in an enhanced SPR signal.
Comment 2: “Authors should provide the synthetic details in the Experimental Section (especially for the synthesis of magnetic nanoparticle).”
Response: The magnetic nanoparticles were purchased from Ruixi Biotech. Co., Ltd. (Xian, China). The functionalization of magnetic nanoparticles has been provided in Part 2.2. The change in the zeta potential (ζ) of magnetic nanoparticles induced by surface modification has been investigated and the result has been discussed in the revised manuscript.
Comment 3: “Authors should provide the high-resolution fluorescence microscopy image of the FITC-NA/BSA/GSH in Figure 1A, which confirm that NA can be anchored on the gold surface.”
Response: We have repeated the experiment several times and found that the fluorescence microscopy image of FITC-NA/BSA/GSH is light green but that of BSA/GSH is completely black, which can be observed by amplifying the figure. The result is understandable since that the FITC-NA proteins were attached onto the gold chip surface through the formation of self-assembly monolayer. We have amplified the image and improved the figure quality in the revised manuscript.
Comment 4: “Authors should provide the comparison of the UV–vis spectra (including MNPs, DNA, Ab1, DNA(Ab1)-MNPs, TDNA(target)/DNA(Ab1)-MNPs, bio-DNA(bio-Ab2)/TDNA(target)/DNA(Ab1)-MNPs) to show the successful vaccination of receptor and functionalized magnetic nanoparticles.”
Response: It is a good suggestion. The DNA probes were immobilized on the surface of carboxylated MNPs through the EDC/NHS-activated amidation covalent coupling reaction. We attempted to characterize the change in the UV–vis spectra of MNPs after surface modification. However, no believable UV–vis results were obtained to confirm the surface modification. Alternately, the change in the zeta potential (ζ) of magnetic nanoparticles induced by surface modification has been investigated and the result has been discussed in the revised manuscript.
Comment 5: “Authors should provide the comparison of the SPR sensorgrams (including Ab1-MNPs, target/Ab1-MNPs, bio-Ab2/target/Ab1, and bio-Ab2/target/Ab1-MNPs) to show the successful formation of the immunocomplex.”
Response: The SPR sensorgrams after injecting Ab1-MNPs, Aβ40/Ab1-MNPs, bio-Ab2/Aβ40/Ab1, and bio-Ab2/Aβ40/Ab1-MNPs to the sensor channels have been added in the revised manuscript.
Comment 6: “In Figure 2(B), authors should add two additional concentrations (in the range between 500~2000 fM) of TDNA to demonstrate practicality of the biosensor in linear quantification range.”
Response: We have added two additional concentrations (1000 and 1500 fM) in the range of 500~2000 fM.
Comment 7: “Please review the language and representation of the whole manuscript to be clear, precise, accurate and focused.”
Response: We have revised the manuscript carefully and asked an English speaker to polish the language.
Reviewer 2 Report
The presented work of the authors is devoted to solving a very urgent problem of creating technologies and sensors for detecting small molecules and ultralow concentrations of analytes. A thorough substantiation of the effectiveness of the method for constructing SPR biosensors by integrating magnetic preconcentration and separation using magnetic nanoparticles is given. The presented results of the development and study of two SPR biosensors for detecting target DNA and beta-amyloid peptide indicate the achieved high sensitivity (multiple signal amplification) and selectivity in the implementation of the proposed approach. At the same time, a relatively high measurement speed is ensured. The manuscript is technically sound with well-supported conclusions and assertions.
This article is well written, it is an interesting experimental work that can be published, but needs the following minor improvements before publication:
1. It is necessary to eliminate the unsatisfactory quality of the images in Fig. 1 (a). There are no visual differences between the two fragments of Fig. 1 (a). Perhaps this is a consequence of importing the file into pdf format. An acceptable way to improve the quality may be color inverting of images.
2. The heading of subsection 3.1 should be changed to avoid ambiguity.
Author Response
We thank the reviewer for his/her positive and constructive comments: “The presented work of the authors is devoted to solving a very urgent problem of creating technologies and sensors for detecting small molecules and ultralow concentrations of analytes. A thorough substantiation of the effectiveness of the method for constructing SPR biosensors by integrating magnetic preconcentration and separation using magnetic nanoparticles is given. The presented results of the development and study of two SPR biosensors for detecting target DNA and beta-amyloid peptide indicate the achieved high sensitivity (multiple signal amplification) and selectivity in the implementation of the proposed approach. At the same time, a relatively high measurement speed is ensured. The manuscript is technically sound with well-supported conclusions and assertions. This article is well written, it is an interesting experimental work that can be published, but needs the following minor improvements before publication:”
Comment 1: “It is necessary to eliminate the unsatisfactory quality of the images in Fig. 1 (a). There are no visual differences between the two fragments of Fig. 1 (a). Perhaps this is a consequence of importing the file into pdf format. An acceptable way to improve the quality may be color inverting of images.”
Response: We have repeated the experiment several times and found that the fluorescence microscopy image of FITC-NA/BSA/GSH is light green but that of BSA/GSH is completely black, which can be observed by amplifying the figure. The result is understandable since that the FITC-NA proteins were attached onto the gold chip surface through the formation of self-assembly monolayer. We have amplified the image and improved the figure quality in the revised manuscript.
Comment 2: “The heading of subsection 3.1 should be changed to avoid ambiguity.”
Response: We have revised the mistake.
Reviewer 3 Report
1. Title can be modified to be short and more attractive
2. How the presently proposed a one-step real-time method for the design of SPR biosensors by integration of magnetic preconcentration and separation satisfy all the requirements like simplicity, rapidity and real-time measurement of SPR biosensors?
3. Magnetic nanoparticles (MNPs) are proposed to be used for receptor modifications to desired capabilities but are the limitations associated with the synthesization as well as use of MNPs are well taken into account.
4. There are many other ways reported in literature towards SERS enhancement sensing activities of substrates. Are these being compared with present methods/techniques for their advantages or disadvantages?
5. High sensitivity and selectivity limits are all relative terms and how these limits are justified presently?
6. All the advantages of present detection technology should be justified.
7. MNPs of the diameter ~100nm used presently seem to very big nanoparticles which will lead to many disadvantages due to this much large size.
8. Materials and methods section (2) gives in very brief the accounts on chemical and materials, functionilzation of MNPs, preparation of SPR chips, SPR analysis which may be difficult for the common reader to understand.
9. Results and discussion section (3), with subsection 3.1 title “Feasibility and feasibility” something is wrong.
10. Why there is such a big difference in Fig.1B and Fig.2A representing all most same parameters and ranges?
11. There are number of research papers highlighting the optical properties of different MNPs from sensitivity purposes which could have been a part of literature review in the present work, like:
· Optical properties simulation of magneto-plasmonic alloys nanostructures, Plasmonics 14 (3), 611-622 (2018)
12. Many units/symbols used in the manuscript are not properly defined/explained.
13. Influence of MNPs on the final optical properties of structures and their uses are not explained.
14. Role of magnetic properties of MNPs are also not discussed.
15. From Table 1 the detection limits of other reported works are still much lower as compared to present work, justification.
16. Manuscript needs revision.
Author Response
Comment 1: “Title can be modified to be short and more attractive.”
Response: We have modified and shortened the title.
Comment 2: “How the presently proposed a one-step real-time method for the design of SPR biosensors by integration of magnetic preconcentration and separation satisfy all the requirements like simplicity, rapidity and real-time measurement of SPR biosensors?”
Response: In the previous reports, antibodies or other receptors are usually anchored on the surface of SPR chip for the capture of target analytes, and then the recognition element-modified magnetic nanoparticles with large molecular weight and high refractive index are delivered onto the chip surface to amplify the SPR signal. The secondary signal amplification can improve the sensitivity and avoid the influence of other components in biological fluids. However, these advantages come at the expense of simplicity, rapidity and real-time measurement due to a greater number of steps in the SPR analysis process. Moreover, the storage of receptors-modified chips requires harsh conditions because of the limited stability of receptors such as antibodies under non-physiological conditions and high temperature, the capture of analytebound MNPs through the hybridization reaction or antibody-antigen interaction requires a long hybridization time and a low flow rate, and the immobilization of receptors on the chip surface may produce a steric hindrance for the capture of analytebound MNPs at the solid-liquid interface. Thus, it is of great significance to develop a general SPR biosensor for the detection of small molecules and ultra-low concentration of analytes by integrating the advantages of one-step real-time detection and multiple signal amplification. Avidin and its analogues including streptavidin (SA) or neutravidin (NA) modified on the solid surface shows high stability, easy controllability and excellent flexibility. Many SA or NA-labeled signal tags or solid supports and biotinylated reagents are commercially available. Herein, we proposed a one-step real-time method for the design of SPR biosensors by integration of magnetic preconcentration and separation. The targets were captured by the receptor-modified magnetic nanoparticles (MNPs) and then the biotinylated recognition elements were attached on the analytebound MNPs to form a sandwich structure. The sandwich hybrids were directly delivered to the neutravidin-modified SPR fluidic channel. The MNPs hybrids were captured by the chip through the neutravidin-biotin interaction, resulting in an enhanced SPR signal.
Comment 3: “Magnetic nanoparticles (MNPs) are proposed to be used for receptor modifications to desired capabilities but are the limitations associated with the synthesization as well as use of MNPs are well taken into account.”
Response: Magnetic nanomaterials (MNPs) have been widely used to enhance the SPR signal due to their advantages of high molecular mass, high refractive index, low production cost and easy synthesis through hydrothermal and co-precipitative methods. We have cited the references and discussed the progress in SPR biosensors by using MNPs as the signal-amplified labels.
Comment 4: “There are many other ways reported in literature towards SERS enhancement sensing activities of substrates. Are these being compared with present methods/techniques for their advantages or disadvantages?”
Response: We have cited the references and discussed the progress in SPR biosensors by using MNPs as the signal-amplified labels.
Comment 5: “High sensitivity and selectivity limits are all relative terms and how these limits are justified presently?”
Response: The sensitivity and selectivity of the method have been investigated. The high sensitivity of the biosensor was conducive to improving the selectivity because the interference from other components in complex biological matrix could be alleviated by dilution of samples.
Comment 6: “All the advantages of present detection technology should be justified.”
Response: We thank the reviewer for his/her comment. The advantages of this method have been discussed with objective descriptions.
Comment 7: “MNPs of the diameter ~100 nm used presently seem to very big nanoparticles which will lead to many disadvantages due to this much large size.”
Response: In the previous reports, MNPs with a diameter from 10 to 1000 nm have been used to amplify the SPR signal because of their advantages of large molecular mass, high refractive index, low production cost and easy synthesis through hydrothermal and co-precipitative methods (Angew. Chem. Int. Ed., 2011, 50, 1175–1178; Coordin. Chem. Rev., 2022, 458, 214424). In this communication, we did not investigate the effect of the size of MNPs on the sensing performances, but proposed a new MNPs-based strategy for the design of SPR biosensors. We greatly appreciate the reviewer’s comment.
Comment 8: “Materials and methods section (2) gives in very brief the accounts on chemical and materials, functionilzation of MNPs, preparation of SPR chips, SPR analysis which may be difficult for the common reader to understand.”
Response: We have revised the manuscript to make it readily understandable.
Comment 9: “Results and discussion section (3), with subsection 3.1 title “Feasibility and feasibility” something is wrong.”
Response: We have revised the mistake.
Comment 10: “Why there is such a big difference in Fig.1B and Fig.2A representing all most same parameters and ranges?”
Response: The concentrations of TDNA were different in the two figures. We have added the information in revised manuscript.
Comment 11: “There are number of research papers highlighting the optical properties of different MNPs from sensitivity purposes which could have been a part of literature review in the present work, like: Optical properties simulation of magneto-plasmonic alloys nanostructures, Plasmonics 14 (3), 611-622 (2018).”
Response: Magnetic nanomaterials (MNPs) have been widely used to enhance the SPR signal due to their advantages of high molecular mass, high refractive index, low production cost and easy synthesis through hydrothermal and co-precipitative methods. We have cited the references to discuss the progress in SPR biosensors by using MNPs as the signal-amplified labels.
Comment 12: “Many units/symbols used in the manuscript are not properly defined/explained.”
Response: We have revised and defined the units/symbols carefully.
Comment 13: “Influence of MNPs on the final optical properties of structures and their uses are not explained.”
Response: Magnetic nanomaterials (MNPs) have been widely used to enhance the SPR signal because of their large molecular mass and high refractive index. We have cited the references and explained the role of MNPs in the SPR biosensor as the signal-amplified labels.
Comment 14: “Role of magnetic properties of MNPs are also not discussed.”
Response: MNPs were used as the signal-amplified labels to enhance the SPR signal because of their large molecular mass and high refractive index. We have discussed the role of MNPs in the revised manuscript.
Comment 15: “From Table 1 the detection limits of other reported works are still much lower as compared to present work, justification.”
Response: We have discussed the results with objective descriptions. In this work, we proposed a new MNPs-based strategy for the design of SPR biosensors. We believe that the analytical performances would be further improved by using magneto-plasmonic alloys nanostructures.
Comment 16: “Manuscript needs revision.”
Response: We thank the reviewer for his/her positive and constructive comments. We have revised the manuscript carefully.
Round 2
Reviewer 3 Report
Authors have successfully answered all the quarries raised by the reviewers and have also incorporated the same in the revised manuscript. The manuscript in its revised form may be accepted.